# NMR-Based Structural Characterization of a Two-Disulfide-Bonded Analogue of the FXIIIa Inhibitor Tridegin: New Insights into Structure–Activity Relationships

**DOI:** 10.3390/ijms22020880

**Published:** 2021-01-17

**Authors:** Thomas Schmitz, Ajay Abisheck Paul George, Britta Nubbemeyer, Charlotte A. Bäuml, Torsten Steinmetzer, Oliver Ohlenschläger, Arijit Biswas, Diana Imhof

**Affiliations:** 1Pharmaceutical Biochemistry and Bioanalytics, Pharmaceutical Institute, University of Bonn, An der Immenburg 4, D-53121 Bonn, Germany; t.schmitz@uni-bonn.de (T.S.); Ajay.PaulGeorge@BioSolveIT.de (A.A.P.G.); Britta.Nubbemeyer@uni-bonn.de (B.N.); charlotte.baeuml@uni-bonn.de (C.A.B.); 2BioSolveIT GmbH, An der Ziegelei 79, D-53757 Sankt Augustin, Germany; 3Institute of Pharmaceutical Chemistry, Philipps University of Marburg, Marbacher Weg 6, 35032 Marburg, Germany; torsten.steinmetzer@staff.uni-marburg.de; 4Leibniz Institute on Aging—Fritz-Lipmann-Institute, Beutenbergstr. 11, D-07745 Jena, Germany; Oliver.Ohlenschlaeger@leibniz-fli.de; 5Institute of Experimental Hematology and Transfusion Medicine, University Hospital Bonn, Sigmund-Freud-Str. 25, D-53127 Bonn, Germany; arijit.biswas@ukbonn.de

**Keywords:** coagulation factor XIIIa, transglutaminase, coagulation cascade, tridegin, peptide inhibitor, cysteine-rich, disulfide bonds, NMR spectroscopy, structure analysis

## Abstract

The saliva of blood-sucking leeches contains a plethora of anticoagulant substances. One of these compounds derived from *Haementeria ghilianii*, the 66mer three-disulfide-bonded peptide tridegin, specifically inhibits the blood coagulation factor FXIIIa. Tridegin represents a potential tool for antithrombotic and thrombolytic therapy. We recently synthesized two-disulfide-bonded tridegin variants, which retained their inhibitory potential. For further lead optimization, however, structure information is required. We thus analyzed the structure of a two-disulfide-bonded tridegin isomer by solution 2D NMR spectroscopy in a combinatory approach with subsequent MD simulations. The isomer was studied using two fragments, i.e., the disulfide-bonded N-terminal (Lys1–Cys37) and the flexible C-terminal part (Arg38–Glu66), which allowed for a simplified, label-free NMR-structure elucidation of the 66mer peptide. The structural information was subsequently used in molecular modeling and docking studies to provide insights into the structure–activity relationships. The present study will prospectively support the development of anticoagulant-therapy-relevant compounds targeting FXIIIa.

## 1. Introduction

Leeches, such as the medical leech *Hirudo medicinalis*, have been used as a biomedical tool from ancient times up to the present day. They were mainly used to remove blood, which should help to detoxify the patient [1]. Because of these bloodsucking properties and the intent to avoid clot formation during blood intake, the leeches’ saliva contains a variety of anticoagulants that can be used for the control and treatment of hemostasis. Over 20 different compounds with antithrombotic activity have already been isolated from the salivary glands of various leeches, showing a direct or indirect influence on the blood coagulation cascade [2,3]. One of these compounds is the well-known thrombin inhibitor hirudin, which is isolated from the leech *Hirudo medicinalis* [4,5]. Based on this natural compound, anticoagulants such as bivalirudine and argatroban have already been successfully developed for the treatment of cardiovascular and thromboembolic diseases [6,7,8]. However, anticoagulants that directly or indirectly inhibit thrombin frequently increase the risk of undesired bleeding, making them unsuitable for many therapeutic applications [9,10]. Since thrombin catalyzes the penultimate step of the blood coagulation cascade—the formation of fibrin polymers starting from fibrinogen—an influence on any of the previously required coagulation factors always indirectly affects thrombin [11]. Therefore, one way to achieve a thrombin-independent impact on the blood coagulation cascade is the inhibition/activation of the final step, the crosslinking of the fibrin polymers catalyzed by FXIIIa [12,13].

FXIIIa is a transglutaminase that forms isopeptide bonds within the fibrin polymer by linking a glutamine side chain and a lysine side chain, providing increased blood clot stability. In this regard, the isopeptide bond formation is catalyzed by a catalytic triad composed of Cys314, His373, and Asp396 and a catalytic dyad (His342, Glu401) located in the active side of FXIIIa [12,14,15,16]. Inhibiting this factor leads to unstable blood clots that are smaller in size and can be easily dissolved by fibrinolysis [17,18,19,20]. Consequently, FXIIIa serves as an interesting target for the development of anticoagulants in order to circumvent the problem of undesired bleeding. So far, several small-molecule inhibitors such as cerulenin and alutacenoic acid A have been investigated for their inhibitory potential against FXIIIa [21,22,23,24]. However, many of these small-molecule inhibitors possess a low selectivity and/or half-life in plasma. Thus, the development of these inhibitors has not been pursued further. Nowadays, the main focus is directed toward peptidic and allosteric FXIIIa inhibitors, such as the peptidomimetic inhibitor ZED3197 and glucosaminoglycan-derived inhibitors, e.g., NSGM 13 (non-saccharide glucosaminoglycan mimetics), introduced by Al-Horani et al. [23,24,25,26]. Furthermore, the peptidic inhibitor tridegin, a 66mer peptide, first isolated from the giant Amazon leech *Haementeria ghilianii* in 1997, has been investigated for its high inhibitory potential against FXIIIa [23,27]. Tridegin belongs to the class of cysteine-rich peptides due to its 6 cysteines within the amino acid sequence. The natural disulfide bond connectivity of tridegin could not be elucidated so far [28,29,30,31], which is due to the fact that a peptide with 6 cysteines has the ability to form 15 different three-disulfide-bonded isomers. So far, final conclusions have not been drawn concerning the folding of tridegin to the bioactive conformation and whether it is folded in a BPTI-like (Bovine pancreatic trypsin inhibitor) fashion, according to which one isomer is preferentially formed, or in a hirudin-like manner, where a maximum of 15 possible three-disulfide-bonded isomers are formed [32,33]. A folding experiment performed by Böhm et al. in 2014 provided first hints regarding the preferentially formed disulfide-bonded isomers of tridegin [30]. Therein, three of the 15 possible three-disulfide-bonded isomers were identified, which all shared the disulfide bond between Cys19–Cys25 and possessed similar activities toward FXIIIa [28,30]. In addition to the three-disulfide-bonded tridegin isomers, several variants lacking the disulfide bond between Cys19–Cys25 were synthesized in a following approach in order to determine the significance of this bond for their inhibitory potential against FXIIIa [29]. The results revealed that the activity against FXIIIa was not lost, despite the missing disulfide bond. Due to the unchanged activity and the improved synthetic outcome, the two-disulfide-bonded tridegin analogues, in particular the isomer B_[C19S,C25S]_ with the disulfide bonds Cys5–Cys37 and Cys17–Cys31, are used as a lead structure for further research on tridegin and its influence on FXIIIa [29].

The aim of the current study is to gain further insight into tridegin’s structure, folding, and interaction with FXIIIa. In order to obtain a more precise understanding of the mode of action against FXIIIa, a structure of the tridegin isomer B_[C19S,C25S]_ was determined by a combination of NMR (nuclear magnetic resonance) spectroscopy and molecular dynamics (MD) simulation studies. This two-disulfide-bonded isomer, B_[C19S,C25S]_, was subsequently applied for molecular docking studies on FXIIIa. The information received from this study was used to refine structure–activity relationships of the tridegin–FXIIIa interaction. Our results may pave the way for further investigations on FXIIIa as well as the optimization/development of lead structures for antithrombotic and thrombolytic therapy.

## 2. Results and Discussion

### 2.1. Peptide Synthesis and Characterization

To elucidate the structure of the two-disulfide-bonded isomer B_[C19S,C25S]_ (Figure 1a) [29], two segments derived from the peptide sequence were synthesized for NMR measurements. The reason for dividing the tridegin isomer into an N-terminal and a C-terminal part was the size of the 66mer peptide and the high number of proline residues within the sequence, which complicate the assignment of the NMR signals and hamper structure calculation. Therefore, a C-terminal fragment (amino acids 38–66) already described by Böhm et al. [31] and a new N-terminal fragment (amino acids 1–37) with the disulfide bonds Cys5–Cys37 and Cys17–Cys31 were produced (Appendix A). For the synthesis of the latter fragment, a solid-phase peptide synthesis using the Fmoc-strategy with acetamidomethyl (Acm)- and trityl (Trt) -protecting groups (Cys5, Cys37 and Cys17, Cys31, respectively) and a subsequent stepwise oxidation in solution was conducted, similar to the strategy used for the earlier described two-disulfide-bonded tridegin analogues [29]. The stepwise oxidation process was performed in a one-pot reaction with iodine (oxidizing agent), whereby the former Trt-protected cysteines, were deprotected during the cleavage of the peptide from the solid support and initially linked, followed by the cleavage of the Acm-protected cysteines (Figure 1b). The selective disulfide bond formation during the synthesis of the two disulfide bonds was controlled by increasing the amount of iodine as described previously [29] and by adjusting the acetic acid concentration. For the disulfide bond formation of the cysteines formerly protected by Trt, the first oxidation was performed in 100% acetic acid, whereas the deprotection of the Acm-protected cysteines and the formation of the second disulfide bond was completed in 70% acetic acid. Due to the higher acid concentration in the first oxidation, the acid-stable Acm-protected cysteines are additionally prevented from unwanted cleavage, thus ensuring the stepwise formation of the disulfide bonds [34]. To confirm the correct disulfide bond connectivity, the N-terminal fragment was digested using chymotrypsin, and the resulting fragments were analyzed subsequently by MS/MS as earlier described [29]. Several disulfide-bonded fragments representative for the disulfide bonds Cys5–Cys37 and Cys17–Cys31 were detected, confirming the correct disulfide bond connectivity of the N-terminal fragment (Figure 1c, Appendix A).

In addition, activity studies of the N-terminal fragment on FXIIIa were performed using the established FXIIIa isopeptidase activity assay [28,29,30]. As already observed in the case of a three-disulfide-bonded N-terminal fragment of tridegin [30], no inhibitory potential against FXIIIa was found. However, as described earlier, it is expected that the N-terminal fragment enhances the binding between tridegin and FXIIIa. In contrast, the C-terminal fragment exhibits inhibitory activity toward FXIIIa [30].

### 2.2. Structure Elucidation by NMR Spectroscopy

The structure of isomer B_[C19S,C25S]_ [29] was determined by combining the two structures of the peptide fragments. First, these structures were elucidated by natural abundance 2D NMR spectroscopy, recording TOCSY (total correlation spectroscopy), NOESY (nuclear Overhauser enhancement spectroscopy), COSY (correlated spectroscopy), and ^13^C-HSQC (heteronuclear single quantum coherence) spectra and calculated by Cyana [35] based on the distance and dihedral angle restraints. Within the signal assignment of the NMR spectra, the spectral resolution of the 2D NMR spectra (TOCSY and NOESY) at a magnetic field of 16.4 T allowed a clear resonance assignment for both peptide fragments (Figure 2, Appendix A).

The TOCSY, COSY, and HSQC spectra of the C-terminal fragment showed a high resolution and well-separated signals, which could be easily associated with the individual spin systems of the 29 amino acids. A subsequent “sequential walk” within the H_N_–H_α_ region of the NOESY and TOCSY spectra (Figure 2a) resulted in a sequence-specific assignment of the signals, despite the difficulty of the occurrence of six prolines. Proline residues lead to an interruption of the sequential walk due to the lack of H_N_. The assignments within the NOESY spectrum were used to solve the solution structures of the C-terminal fragment via Cyana [35]. The NOESY spectrum already revealed few correlations between amino acids that were separated by more than two amino acids within the sequence. This resulted in a very small number of distance restraints for the structure calculation, which was reflected in a flexible structure of the C-terminal fragment (Figure 3, Appendix A) with a backbone root mean square deviation (RMSD) of 8.54 ± 2.83 Å.

The NMR spectra of the N-terminal fragment contained more signals due to the higher number of amino acids (37 amino acids), which were successfully attributed to the amino acid positions by assigning the amino acid spin system and subsequent sequential walk within the H_N_–H_α_ range (Figure 2b), as it was already performed for the C-terminal fragment. The two disulfide bonds as well as possible secondary structure elements within the N-terminal fragment result in a more rigid structure compared to the C-terminal fragment, enabling NOESY correlations between distant amino acids within the peptide sequence. In this context, NOESY correlations between the amino acids Trp8–Gln36, Pro15–Phe33, Cys17–Phe33, Arg16–Tyr30, Trp18–Cys31, and Ser19–Tyr30 confirmed the disulfide bond connectivity Cys5–Cys37 and Cys17–Cys31 of the N-terminal fragment, which was already verified by the aforementioned MS/MS analysis of the chymotryptic digest. The numerous NOESY correlations led to a higher set of distance restraints, which, in addition to the dihedral angle restraints, were included in the calculation of the structure via Cyana [35]. In contrast to the C-terminal fragment, three secondary structure elements stabilized by the disulfide bonds Cys5–Cys37 and Cys17–Cys31 were identified within the N-terminal fragment (RMSD of residues 5–37: 1.20 ± 0.39 Å), e.g., one helix (Glu24–Tyr30) and two β-turns (His9–Ile12, Trp18–Ala21) (Figure 3, Appendix A). Especially the fact that the two secondary structure elements in the region Trp18–Tyr30 result in a close proximity of the serine residues at position 19 and 25 is remarkable. This may indicate that the disulfide bond between Cys19–Cys25, which occurs in the three-disulfide-bonded tridegin isomers [30], is formed as a consequence of the secondary structure elements. Regarding this, it may be concluded that during the folding process, this bond is only established after the formation of the secondary structure elements.

To determine the isomer B_[C19S,C25S]_ structure [29], the distance and dihedral angle restraints of the N- and the C-terminal fragments were combined and the complete structure ensemble (100 structures, Figure 4) of the tridegin variant was calculated using Cyana [35]. Thereby, the determined secondary structure elements of the N-terminal fragment as well as the flexibility of the C-terminal fragment were preserved. The possibility of the C-terminal fragment (residue 38–66) to undergo large conformational changes was characterized by an RMSD of 14.14 ± 2.76 Å for the combined isomer B_[C19S,C25S]_ structure [29] with reference to the lowest energy structure (Figure 4a). Additionally, within the structure ensemble of isomer B_[C19S,C25S]_ [29] a massive proportion of this observed variation (12.15 ± 3.27 Å) resulted from the flexible C-terminal fragment (residues 38–66) while the N-terminal fragment alone (residues 1–37) had a backbone RMSD of 3.77 ± 1.42 Å. However, since the calculated structure ensemble of isomer B_[C19S,C25S]_ [29] is only a combination of the independently resolved NMR structures of the N- and C-terminal fragment, the structures were further investigated via molecular dynamics simulations.

### 2.3. MD-Based Analysis of Isomer B_[C19S,C25S]_

For the investigation of the aforementioned isomer B_[C19S,C25S]_ structure [29], the model with the lowest energy from the 100-member NMR ensemble of the calculated isomer B_[C19S,C25S]_ structures [29] was subjected to a 300-ns-long all-atom molecular dynamics. The prime motivation behind this effort was to produce a solvent-equilibrated ensemble of the isomer B_[C19S,C25S]_ structure [29], which was originally produced by combining the independently resolved fragments (N-terminal and C-terminal fragments, as described above) determined by NMR spectroscopy. Additionally, the final structure from this MD simulation was also used as an input for molecular docking experiments on the crystal structure of FXIIIa (PDB ID: 4KTY) [16].

As already discussed in Section 2.2, analyses conducted of the NMR ensemble of the combined (N-terminal and C-terminal fragments) isomer B_[C19S,C25S]_ structure [29] (Figure 4a) strongly indicated that the elongated C-terminal segment could potentially undergo large conformational changes upon further equilibration to attain its fully folded conformation. It was therefore clear that, prior to molecular docking experiments, MD-based refinement of this structure was necessary. To this end, a 300-ns-long all-atom MD simulation was performed on the NMR structure with the lowest energy (Figure 4b), and the resulting trajectory was analyzed for structural changes over the course of the simulation. A large conformational rearrangement marked by an 18.50 Å change in backbone RMSD was observed within the first 15 ns of the 300 ns simulation (Figure 4c). This was characterized by a rapid collapse of the flexible C-terminal fragment (residues 38–66) around the disulfide-bonded N-terminal segment (residues 1–37). Remarkably, after this initial dramatic change, the remainder of the simulation saw the structure steadily equilibrate deviating only by an average of 3.33 ± 0.47 Å with respect to the structure at 15 ns. The large deviation between the NMR starting structure and the final simulation structure can clearly be explained due to the fact that, in the NMR structure, the N- and C-terminal segments were resolved as separate entities and were combined as a final step to create the full-length isomer B_[C19S,C25S]_ structure [29]. We believe that our simulation captures the full extent of conformational changes the structure would have undergone if the N-terminal and C-terminal segments could possibly have been resolved as a single entity via NMR spectroscopy and that the structures obtained from any part of the trajectory after the initial 15 ns would surely fall within the low-energy conformations of isomer B_[C19S,C25S]_ [29] in its physiological environment. To further confirm this, we show that the conformational ensemble for the entire structure produced via the MD simulation (containing 2850 snapshots) has an RMSD of 3.33 ± 0.47 Å, which is comparable to the RMSD (3.77 ± 1.42 Å) of the 100-member NMR ensemble of the N-terminal segment of isomer B_[C19S,C25S]_ [29]. We also note that the α-helical motif seen in the NMR starting structure (Glu24–Tyr30) was consistently found through the production phase of the MD simulation. The final snapshot of this 300 ns simulation was used as an input structure for subsequent molecular docking simulations. A video of this MD simulation is provided as supplementary content. Structural alignments of the final snapshot of the simulation with a previously computationally modeled version of the same peptide [29] using the MUSTANG algorithm [36] revealed that although the general fold was preserved, the NMR-based model (current study) varied from the earlier computational model (Figure 5b–d) [29]. This is understandable based on the fact that the earlier model was produced by mutating its three-disulfide-bonded counterpart, which again had its origins in computational modeling. Interestingly though, the parts of the C-terminal segment, seemed to align well against each other (Figure 5b–d). The current produced model has the clear advantage that its origin is from NMR spectroscopy. As a means to confirm that this MD-derived structure from the 300 ns simulation was reproducible, a second, independent 1000-ns-long simulation was conducted. The equilibrated structures from this longer simulation aligned well with the structure produced by the 300 ns simulation in this study (Appendix A).

### 2.4. Molecular Docking of Full-Length Tridegin Variant on FXIIIa

Docking of the full-length model of isomer B_[C19S,C25S]_ [29] (derived from the final 300 ns simulation snapshot) (Figure 5a) onto FXIIIa resulted in one pose amongst the top-ten docking poses that satisfies the inhibitory characteristic of tridegin (i.e., it binds next to the catalytic site) (Appendix A). All remaining docks cluster around the β-barrel domain of FXIIIa, which is located significantly far from the FXIIIa catalytic site, and are therefore irrelevant to our analysis. On an atomic/residual level, the important docking pose of isomer B_[C19S,C25S]_ [29] shows the C-terminal region (residues 53–60) participating in several non-covalent interactions in and around the catalytic site (Figure 6, Appendix A). The original non-proteolytically activated crystal structure from Stieler et al. [16] presents a hydrophobic tunnel at the active site formed by the aromatic interactions between the Trp279 and Trp370 residues of FXIIIa (Figure 6). The two ends of this tunnel serve as the entry points for the FXIIIa substrates, lysine and glutamine, through which they reach the catalytic Cys314 at the bottom of the cavity that the tunnel is formed on top of. The original non-proteolytically activated crystal structure of FXIIIa (PDB ID: 4KTY [16]) was stabilized using the FXIIIa inhibitor ZED1301, which irreversible bound to the catalytic Cys314, thereby occupying the hydrophobic tunnel and preventing access for competitive substrate [16]. The mode of inhibition for the full-length model of isomer B_[C19S,C25S]_ [29] is very similar to that of ZED1301 since residues from the C-terminal region not only participate in hydrophobic interactions with FXIIIa residues that would form the hydrophobic tunnel, but some tridegin residues such as Arg53 and Phe57 also stabilize the C-terminal region by interacting with the critical Trp279 and Trp370 FXIIIa residues (Appendix A). Therefore, the C-terminal region appears to competitively occupy the substrate-entry hydrophobic tunnel, thereby preventing substrate access to the catalytic Cys314 similar to ZED1301. An improvement compared to the inhibitor ZED1301 might be that the model structure of isomer B_[C19S,C25S]_ [29] also potentially blocks the catalytic dyad of FXIIIa (His342, Glu401) which is responsible for incorporating the substrate lysine into the hydrophobic tunnel. This is suspected since the inhibitor ZED1301 does not reach the catalytic dyad, whereas the tridegin analogue rests on the residues His342 and Glu401. In addition to the ZED1301 comparison, the docking results of the NMR-based model structure was aligned to a previously reported docking with the computationally modeled version of the same peptide (Appendix A) [29]. In this regard, both dockings showed a similar binding into the hydrophobic tunnel of FXIIIa, despite the overall structural variations between the two (Figure 5b–d). However, there is one significant difference between the two complexes, which is the orientation of the C-terminal part within the hydrophobic pocket. While the NMR-based structure guide through the hydrophobic tunnel showed that the C-terminus of the peptide is on the Q-site (site of the tunnel where glutamine normally binds) of FXIIIa, the C-terminus of the previously reported structure [29] was observed on the K-site (site where lysine binds) close to the catalytic dyad. The differences in the orientation can be attributed to two main things: (a) firstly the differences in the N-terminal structure (residue 1–37) in both isomer B_[C19S,C25S]_ models [29], whose role in this interaction is to orient the C-terminal region within the FXIIIa hydrophobic pocket, and (b) methodological differences in the implementation of docking between the current and previous study. The similarity in binding location though further underlines the importance of the C-terminal fragment in binding and subsequent inhibition of FXIIIa.

Although the Arg16 residue from the tridegin isomer B_[C19S,C25S]_ N-terminal part [29] of the NMR-based model also appears to interact with FXIIIa in this docking pose, the effect of the N-terminal region (residue 1–37) might only be complimentary. The interaction may only play a role in the initial establishment of contact between isomer B_[C19S,C25S]_ [29] and FXIIIa but is unlikely to play any role in the subsequent inhibition. This was confirmed by a docking were only the N-terminal fragment was docked to FXIIIa. Similar to isomer B_[C19S,C25S]_ [29], when we dock only the NMR-based N-terminal fragment structure of the tridegin analogue on FXIIIa, one docking pose amongst top-ten docked close to the catalytic site. However, unlike the full-length isomer B_[C19S,C25S]_ [29] model, the N-terminal fragment structure binds only superficially on top of the catalytic site (Appendix A). In this docking pose, neither the hydrophobic tunnel nor any hydrophobic access is disrupted, which is also a hint for the reduced influence of the N-terminal part on the inhibition to FXIIIa.

## 3. Materials and Methods

### 3.1. Materials

Fmoc-amino acids were purchased from Orpegen Peptide Chemicals (Heidelberg, Germany) and Novabiochem (Schwalbach, Germany). HBTU, resins, and further chemicals for solid-phase peptide synthesis, including reagent-grade N-methylmorpholine, piperidine, trifluoroacetic acid (TFA), and N,N-dimethylformamide (DMF) were purchased from IRIS Biotech (Marktredwitz, Germany), Sigma-Aldrich Chemie GmbH (Munich, Germany), Alfa Aesar (Karlsruhe, Germany), Abcr GmbH (Karlsruhe, Germany), VWR International (Darmstadt, Germany), and FLUKA Chemika (Seelze, Germany). Solvents (acetonitrile, water, methanol, ethyl acetate, diethyl ether) and chemicals (iodine, acetic acid) used for peptide purification and oxidation were obtained from VWR International (Darmstadt, Germany) and Fisher Scientific (Schwerte, Germany).

### 3.2. Peptide Synthesis and Purification

The synthesis of the linear peptide precursors according to an automated standard Fmoc SPPS protocol using a ResPep SL peptide synthesizer from Intavis Bioanalytical Instruments GmbH (Cologne, Germany) as well as the subsequent peptide cleavage from the resin and the purification step via RP-HPLC were carried out as previously described [29].

The purity of the linear peptides was >95%, which was assessed by analytical RP-HPLC on a Shimadzu LC-20AD system equipped with a Vydac 218TP column (C18, 250 × 4.6 mm, 5 µm particle size, 300 Å pore size). The gradient elution system contained 0.1% TFA in water (eluent A) and 0.1% TFA in acetonitrile (eluent B). Peptide elution was achieved with a gradient of 20% to 50% eluent B in 30 min and a flow rate of 1 mL/min. The peptides were detected at λ = 220 nm.

### 3.3. Selective Oxidation of the N-Terminal Fragment

The selective oxidation of the two-disulfide-bonded N-terminal peptide Lys1–Cys37 proceeded in a one-pot reaction slightly modified to the already described protocol [29]. In brief, the linear precursors (1 eq.) were dissolved in 100% acetic acid to yield a final concentration of 0.05 mM. The oxidation of the first disulfide bond with 1.1 eq. iodine (0.1 M in methanol) was stirred at room temperature and under argon atmosphere. After 15 min, 13.9 eq. iodine (0.1 M in methanol) and water was added to the reaction mixture in order to decrease the acetic acid concentration to 70%. The reaction was stirred for a further 43 h (second oxidation). The oxidation reaction was stopped by the removal of iodine by extraction with the same volume of ethyl acetate (three times). The peptide-containing aqueous solutions were combined, freeze-dried, and purified via RP-HPLC.

### 3.4. Peptide Characterization and Determination of Disulfide Connectivity

Peptide characterization was achieved by means of analytical RP-HPLC (see above for the linear precursor), mass spectrometry, and amino acid analysis.

The peptide concentration as well as the amino acid composition were analyzed using an LC 3000 system from Eppendorf-Biotronik (Hamburg, Germany). Peptide hydrolysis was carried out in 6 N HCl at 110 °C in sealed tubes for 24 h. Subsequently, the hydrolyzed peptides were dried in a vacuum concentrator and redissolved. The respective amino acid concentrations were determined by comparison with an amino acid standard solution (Laborservice Onken, Gründau, Germany).

Peptide masses were determined using matrix-assisted laser desorption/ionization (MALDI) mass spectrometry. As matrix 2,5-dihydroxyacetophenone (2,5-DHAP) was used according to the manufacture’s guideline (Bruker Daltonics, Bremen, Germany). MALDI mass spectra were produced on an UltrafleXtreme instrument (Bruker Daltonics, Bremen, Germany).

The disulfide connectivity of the N-terminal fragment was determined as described recently [29,30]. Therefore, a micrOTOF-Q III device (Bruker Daltonics, Bremen, Germany) was used to measure electrospray ionization (ESI) mass and tandem-mass spectra.

### 3.5. Enzyme Activity Assay

The inhibitory potential of the N-terminal tridegin fragment toward FXIIIa was performed in an FXIIIa isopeptidase activity assay using the fluorogenic substrate H-Tyr(3-NO_2_)-Glu(NH-(CH_2_)_4_-NH-Abz)-Val-Lys-Val-Ile-NH_2_ as described previously [30].

### 3.6. NMR Spectroscopy and Structure Prediction

The natural abundance NMR experiments for the ^1^H, ^15^N, and ^13^C chemical shift assignments were performed at 293 K in 50 mM sodium phosphate pH 6.3 (95% H_2_O/5% D_2_O) on a Bruker Avance III HD 700 MHz Cryo spectrometer. The peptides were measured at a concentration of 3 mM, and the backbone as well as the side chain atoms were assigned via a combination of 2D [^1^H, ^1^H]-TOCSY, [^1^H, ^1^H]-NOESY, [^1^H, ^1^H]-COSY, and [^1^H, ^13^C]-HSQC spectra using water suppression. The spectra were processed with TOP SPIN 4.0.6 (Bruker) and analyzed using CcpNmr Analysis (Collaborative Computing Project for NMR). Distance constraints were extracted form [^1^H,^1^H]-NOESY spectra acquired with a mixing time of 150 ms and a recycle delay of 1.5 s. Upper-limit-distance constraints were calibrated according to their intensities in the NOESY spectra. To calculate the structure of the fragments based on the chemical shift data the program Cyana [35] was used. Cyana [35] was also used to calculate a hypothetical structure of the full-length tridegin analogue out of the combined distance and dihedral restraints from the NMR experiments of the N- and C-terminal fragments. The 100 structures with the lowest energies were selected to represent the NMR-solution structures.

### 3.7. Molecular Dynamics (MD) Simulations

Molecular dynamics (MD) simulations in this study were carried out using the Gromacs 2018 package [37,38]. The first model of the 100-member NMR ensemble of the full-length model of isomer B_[C19S,C25S]_ [29] was used as the starting structure of the 300-ns-long simulation. The peptide was placed in the center of a cubic simulation cell, with the edges of the cube separated by at least 12 Å from all atoms of the peptide. The peptide was solvated using the TIP3P water model [39] with additional Na+ and Cl− counter ions added to achieve a physiological salt concentration of 0.9%, while maintaining a zero net charge on the entire system. The solvated system was subjected to 5000 steps of steepest-descents energy minimization with the Amber-ff14sb [40] force field, which was used to describe atomic motions for all parts of the simulation. The energy-minimized system was first subjected to a temperature equilibration in the NVT (constant number of atoms, volume, temperature) ensemble for 2 ns, with temperature maintained by the velocity-rescaled variant of the Berendsen thermostat at 300 K [41]. Subsequently a 2 ns simulation to equilibrate the pressure of the system at 1 atm was conducted in the NPT (constant number of atoms, pressure, temperature) ensemble aided by the Parrinello–Rahman barostat [42,43]. During both the temperature and pressure equilibration runs, all heavy atoms were position-restrained by the LINCS algorithm [44]. The production run was conducted for 300 ns using a 2 fs timestep. Long-range interactions were cutoff at 10 Å, and the electrostatics were described by the particle-mesh Ewald method [45,46]. Periodic boundary conditions were employed in the simulations, the effects of which were adjusted prior to conducting analyses on the trajectory. Trajectory analysis, namely the backbone root mean square deviation (RMSD) computation was conducted in VMD 1.9.3. [47], which was also used in the creation of molecular graphics. Snapshots were written at a 100 ps interval to disc, resulting in a total of 3000 snapshots collected for analysis, from this simulation.

### 3.8. Blind Docking Studies on the FXIIIa Crystall Structure (PDB ID: 4KTY)

The full-length model of isomer B_[C19S,C25S]_ [29] was used for docking onto the modified and simulation-equilibrated crystal structure of FXIIIa originally derived from the source PDB ID: 4KTY [16]. The PDB structure used as a receptor for docking was a simulation-averaged structure of the non-proteolytically activated FXIIIa, which had been modified, i.e., missing regions/loop filled, all heteroatoms/water molecules removed, and subjected to classical molecular dynamic simulation. The details of this simulation have been reported earlier [15]. The full-length model obtained as the final snapshot of the 300 ns MD simulation (described in Section 3.7) was used as the ligand for the docking simulation. Docking was performed on the Hdock server (http://hdock.phys.hust.edu.cn/) in a blind fashion and under the default conditions of the server [48]. The Hdock server is based on a hybrid algorithm of template-based modeling and ab initio free docking and is currently ranked favorably amongst the topmost automated docking servers in the recent CASP (critical assessment of protein structure prediction) competitions [49]. Only the top-ten docking poses generated by the server were closely inspected. The docking pose that strongly agreed with the experimental data was subjected to a docking refinement protocol on the Haddock 2.2 webserver (https://milou.science.uu.nl/services/HADDOCK2.2/haddockserver-refinement.html) that allows for flexible refinement in explicit solvent of the docked complex [50]. The model with the best Haddock score from the output clusters generated out of the refinement protocol was finally inspected in terms of interatomic interactions on the protein interaction calculator webserver (http://pic.mbu.iisc.ernet.in/) [51]. Similarly docking was also performed with the energetically minimum NMR structure of the well-defined N-terminal fragment as the ligand and the above-described FXIIIa structure as the receptor.

## 4. Conclusions

In summary, we have experimentally determined for the first time, the structure of a two-disulfide-bonded tridegin variant. This structure contained a flexible C-terminal segment and a more rigid N-terminal part which showed distinct secondary structure elements besides being stabilized by the two disulfide bonds Cys5–Cys37 and Cys17–Cys31. The cysteine-rich N-terminal fragment also shows that the two serine residues Ser19 and Ser25, which are substituted by the cysteines Cys19 and Cys25 in the three-disulfide-bonded tridegin isomers, were in close proximity to a helix and a β-turn secondary structure element, which suggests that the secondary structure elements are responsible for the formation of this specific disulfide bond. Therefore, it can be assumed for the disulfide bond Cys19–Cys25 in the three-disulfide-bonded tridegin isomer that it is generated after the formation of the secondary structure elements within the folding process. With regard to the structure–activity relationships, it can be assumed that the low structural influence of the disulfide bond results in an already experimentally described comparable binding behavior and inhibitory potential to FXIIIa for the three-disulfide-bonded isomer B [28] and the two-disulfide-bonded isomer B_[C19S,C25S]_ [29].

The full-length structure of isomer B_[C19S,C25S]_ [29] generated from the NMR-based results of the N-terminal and C-terminal fragment was subsequently refined by MD simulations and used for molecular docking to FXIIIa. Thereby, it could be demonstrated that the isomer B_[C19S,C25S]_ [29] binds in close proximity to the active site resulting in a blockage of the catalytic Cys314 for substrates as described earlier [29]. The isomer B_[C19S,C25S]_ [29], especially the C-terminal region, penetrates into the hydrophobic tunnel where the active site of FXIIIa is located and competitively inhibits the transglutaminase functionality of FXIIIa by blocking the catalytic triad and catalytic dyad. The interaction with the hydrophobic tunnel of FXIIIa demonstrates how important the flexible C-terminal region is for the inhibition mode of isomer B_[C19S,C25S]_ [29] and that the rigid N-terminal segment only plays a minor role for the inhibition by enhancing the binding to FXIIIa.

In conclusion, the deeper insight into the structure–activity relationships of the FXIIIa inhibitor tridegin from the docking studies will prospectively support the future development of anticoagulant-therapy-relevant compounds targeting FXIIIa.

## Figures and Tables

**Figure 1 ijms-22-00880-f001:**
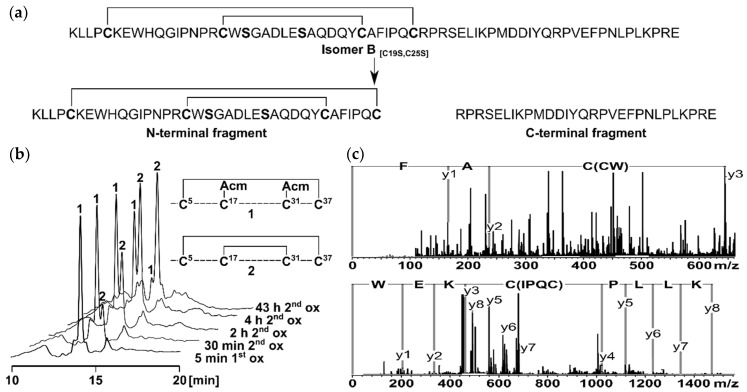
(**a**) Derivation of the two fragments from the two-disulfide-bonded tridegin isomer B_[C19S,C25S]_ [29]. The N-terminal fragment consists of amino acid 1–37 and contains the disulfide bonds Cys5–Cys37 and Cys17–Cys31. The C-terminal fragment is composed of amino acids 38–66 without a cysteine residue; (**b**) HPLC elution profiles of the stepwise oxidation strategy applied for the synthesis of the N-terminal fragment. During the first oxidation (1.1 eq. iodine, 100% AcOH) the first disulfide bond between Cys5 and Cys37 was formed (1). Subsequently the second disulfide bond between Cys17 and Cys31 was built (2) by increasing the amount of iodine (15 eq.) and adjusting the concentration of AcOH (70%); (**c**) MS/MS analysis of the N-terminal fragment after digestion with chymotrypsin. The disulfide bond connectivity was confirmed by the characteristic MS/MS fragments C(CW) AF and KLLP C(IPQC) KEW.

**Figure 2 ijms-22-00880-f002:**
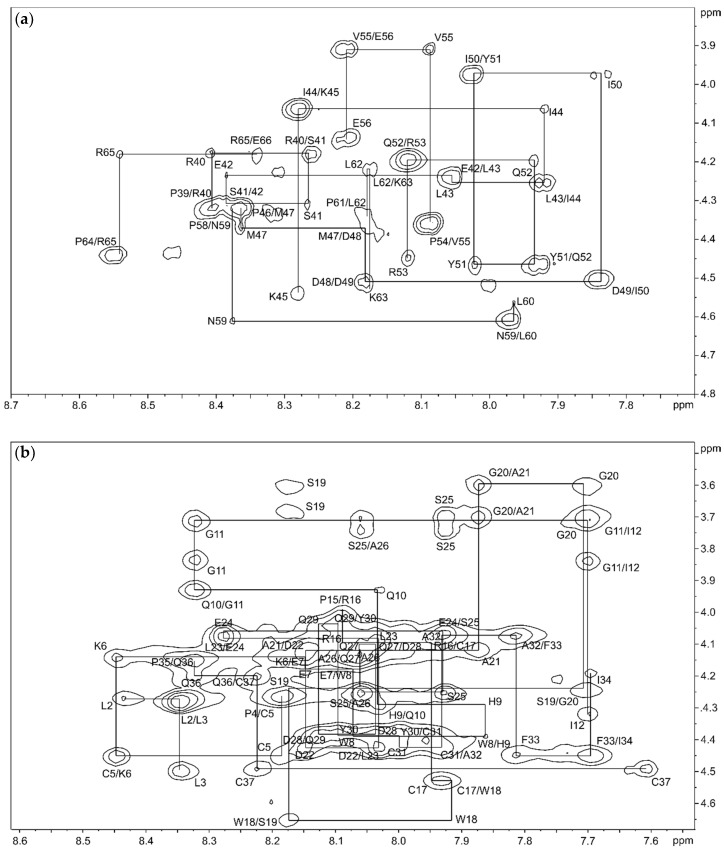
Backbone H_N_–H_α_ region of the 150 ms NOESY (nuclear Overhauser enhancement spectroscopy) spectrum of the C-terminal fragment (**a**) and the N-terminal fragment (**b**) in 95% H_2_O/5% D_2_O with backbone connectivities labeled.

**Figure 3 ijms-22-00880-f003:**
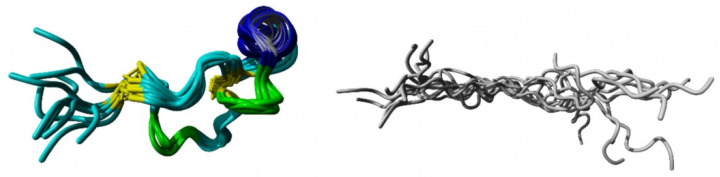
NMR structure of the rigid N-terminal fragment (**left**) and the flexible C-terminal fragment (**right**) presented as ensembles of the 10 structures with the lowest energies. grey: flexible C-terminal fragment; yellow: disulfide bonds Cys5–Cys37 and Cys17–Cys31; blue: helix; green: β-turns.

**Figure 4 ijms-22-00880-f004:**
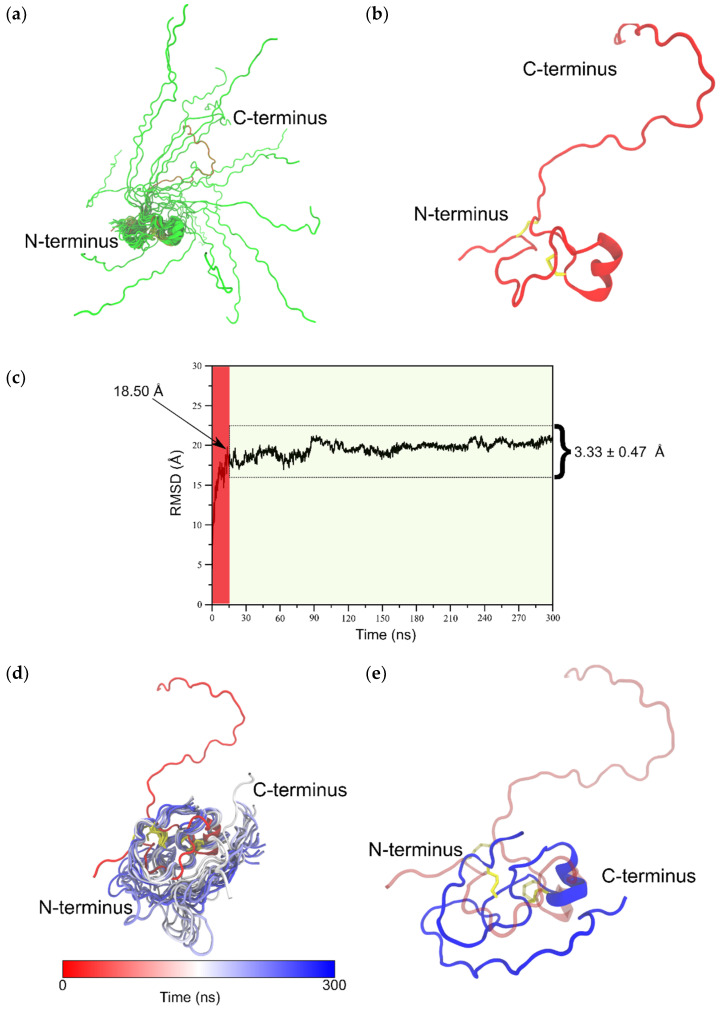
(**a**) Figure depicts 20 structures (green cartoons) from the 100-member NMR ensemble of full-length isomer B_[C19S,C25S]_ [29] aligned on the N-terminal residues (1–37). The lowest energy model 1 is shown as red cartoons. (**b**) The lowest energy model from the NMR ensemble of isomer B_[C19S,C25S]_ [29] used as input in the 300 ns molecular dynamics (MD) simulation. (**c**) Backbone root mean square deviation (RMSD) trace from the 300 ns MD simulation of isomer B_[C19S,C25S]_ [29] with respect to its starting structure. The initial 15 ns of this simulation is considered as the equilibration phase (shaded red) and the remainder of the simulation (shaded green) is considered the production phase of the simulation. (**d**) Figure presents 20 equidistant structures from between 150 and 300 ns of the simulation as an ensemble (white to blue gradient cartoons) with the starting structure (red cartoon) shown as an indicator of the extent of structural change especially at the C-terminal (residues 38–66). (**e**) The final snapshot of the 300 ns MD simulation (blue cartoon) shown along with the starting structure (red transparent cartoon).

**Figure 5 ijms-22-00880-f005:**
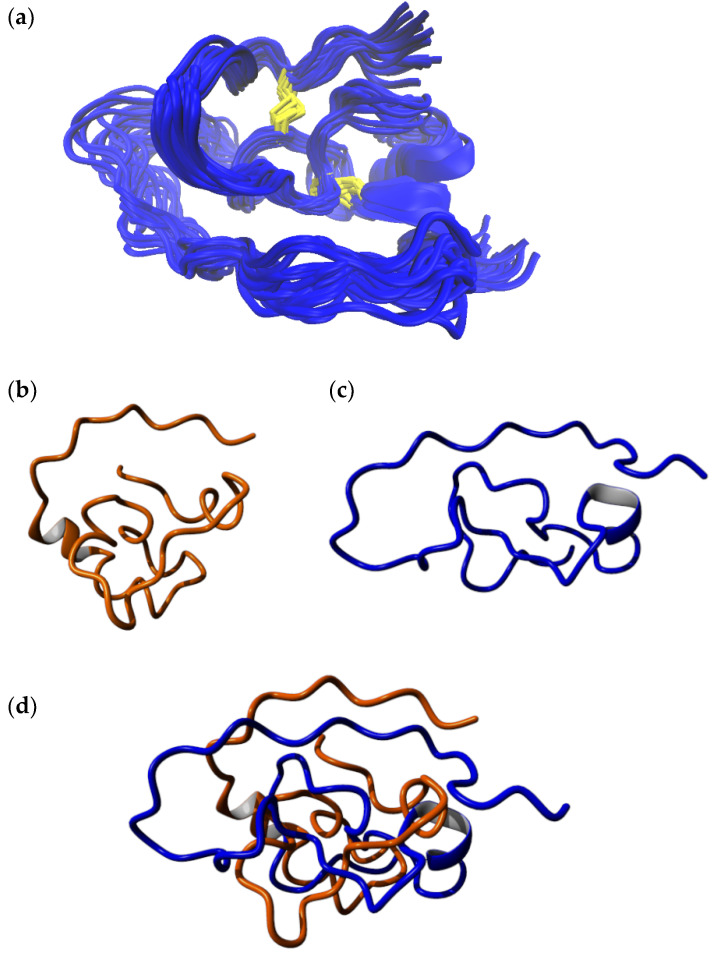
(**a**) The final 20 snapshots of the 300 ns MD simulation of the full-length model of isomer B_[C19S,C25S]_ [29]. (**b**) Computationally derived structure of the same peptide analogue from an earlier study [29]. (**c**) Final snapshot from the 300 ns simulation from the current study. (**d**) Structural alignment of the final structure from the 300 ns MD simulation from the current study (blue) with the computationally derived structure of the same peptide (orange) from an earlier study [29]. The alignment indicates that the two models essentially have varying structures while the underlying fold is preserved. The computationally derived structure from earlier work has a more compact fold owing to the fact that it was created based on its three-disulfide-bonded parent, where residues Cys19 and Cys25 were linked by a disulfide bond. A fair alignment between the trailing end of the C-terminal segment is observed. The alignment was carried out using the MUSTANG algorithm [36].

**Figure 6 ijms-22-00880-f006:**
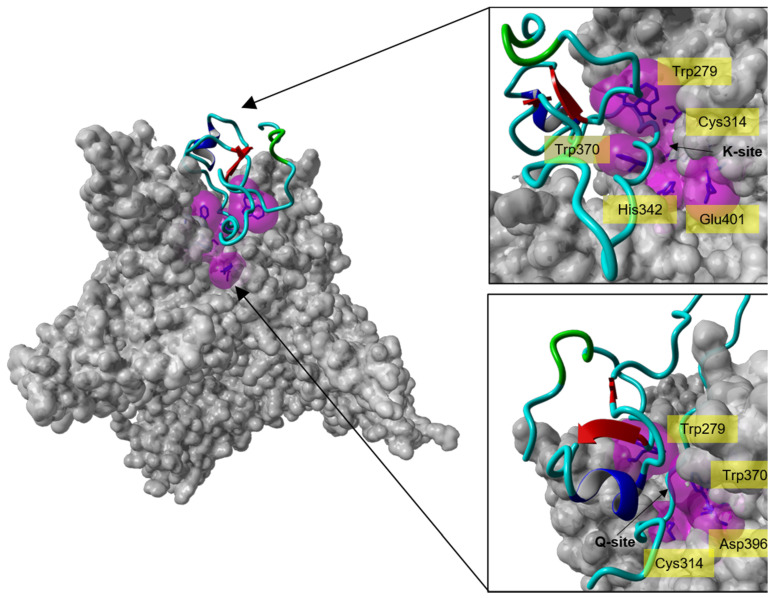
Molecular docking analysis of the interaction of isomer B_[C19S,C25S]_ [29] with the catalytic site of FXIIIa. **Left**: Crystal structure of the calcium-activated FXIIIa-monomer (PDB ID: 4KTY [16], grey surface) docked with one of the top-ten-ranked conformations of isomer B_[C19S,C25S]_ [29] (cyan ribbon), which interacts with the catalytic site of FXIIIa. **Right**: Zoom-in from different directions (lysine access site (**top**, K-site) and glutamine access site (**bottom**, Q-site)) of the FXIIIa active site interacting with isomer B_[C19S,C25S]_ [29]. The active site residues of FXIIIa (Cys314, His373, Asp396, Trp279, and Trp370) are shown as blue sticks enclosed by their molecular surfaces (violet translucent surface).

## Data Availability

The data presented in this study are available on request from the corresponding author.

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
