# Peer review of "NMR-Based Structural Characterization of a Two-Disulfide-Bonded Analogue of the FXIIIa Inhibitor Tridegin: New Insights into Structure–Activity Relationships"

_ijms, 2021, doi:10.3390/ijms22020880_

Round 1
Reviewer 1 Report
The results presented in the article “NMR-based structural characterization of a 2-disulfide bonded analogue of the FXIIIa inhibitor tridegin: New insights into structure-activity relationships” are of scientific interest. The choice of objects and methods believed to be reasonable and relevant.
The authors are studying fragments of tridegin (N-terminal (1-37), C-terminal (38-66)) in order to avoid correlation problems in NMR studies of the full peptide. The restrictions obtained from NMR experiments (ex: dihedral angles, distances) of tridegin fragments were further used to calculate the spatial structure of full peptide. This approach has the right to exist, because the authors approve the resulting structure using molecular dynamics. I do not question the accuracy of the calculations. As a possible improvement to the article, I would advise the authors to record the NMR spectra of tridegine and compare them with the spectra of fragments (N-terminal (1-37) and C-terminal (38-66)). If there is an overlap of signals, then this will become legitimacy proof for using the restrictions of fragments when calculating the spatial structure of full peptide. This recommendation does not call into question the results presented in the article and is of a recommendatory nature.
I believe that the presented work can be recommended for publication in the International Journal of Molecular Sciences with minor revision:
- In the presentation of the work, it is necessary to correct Figure 5b, it is not informative. I would encourage authors to submit structures separately.
Author Response
Dear Silverdew Mo,
Please find enclosed our revised manuscript (ID: ijms-1068812) entitled “NMR-based structural characterization of a 2-disulfide bonded analogue of the FXIIIa inhibitor tridegin: New insights into structure-activity relationships” by Schmitz et al. for the International Journal of Molecular Sciences.
We like to thank the reviewers for their useful comments and appreciate the opportunity to address the questions and comments raised about our manuscript. As you will see, we have used the time to answer all concerns and questions. Thus, we feel that we have addressed all the points raised.
In response to the comments/questions, we have done the following to make our responses easy to read and to identify:
- We included all the reviewer comments below.
- Our responses are included immediately after each comment under “Author comment”.
- Changes in the manuscript can be retraced as track changes in the “Supporting Information for Review Only” files of the main text and the supporting information which is submitted in addition to the main “Resubmission” files where all track changes have been accepted.
Reviewers' comments:
Reviewer 1:
The results presented in the article “NMR-based structural characterization of a 2-disulfide bonded analogue of the FXIIIa inhibitor tridegin: New insights into structure-activity relationships” are of scientific interest. The choice of objects and methods believed to be reasonable and relevant.
The authors are studying fragments of tridegin (N-terminal (1-37), C-terminal (38-66)) in order to avoid correlation problems in NMR studies of the full peptide. The restrictions obtained from NMR experiments (ex: dihedral angles, distances) of tridegin fragments were further used to calculate the spatial structure of full peptide. This approach has the right to exist, because the authors approve the resulting structure using molecular dynamics. I do not question the accuracy of the calculations. As a possible improvement to the article, I would advise the authors to record the NMR spectra of tridegine and compare them with the spectra of fragments (N-terminal (1-37) and C-terminal (38-66)). If there is an overlap of signals, then this will become legitimacy proof for using the restrictions of fragments when calculating the spatial structure of full peptide. This recommendation does not call into question the results presented in the article and is of a recommendatory nature.
Author comment: We like to thank the reviewer for the careful evaluation of our manuscript and the positive statement. We are also thankful for the comment to compare the fragment spectra with the NMR spectrum of isomer B[C19S,C25S]. However, since we were not able to carry out any labelling experiments due to the complex synthesis strategy, the NMR spectra at natural abundance of tridegin isomer B[C19S,C25S], which have already been measured, are very complex and therefore very difficult to compare with the other spectra. In addition, the chemical environment of the individual molecules are variable, which induces slight changes of the chemical shifts in the individual spectra and makes a perfect overlay of the spectra difficult.
I believe that the presented work can be recommended for publication in the International Journal of Molecular Sciences with minor revision:
- In the presentation of the work, it is necessary to correct Figure 5b, it is not informative. I would encourage authors to submit structures separately.
Author comment: We understand that Figure 5b might look somewhat confusing since the alignment of the two structures is not perfect. Based on the reviewer’s recommendation, we have now presented the structures separately before the alignment is shown (see optimized Figure 5 in the manuscript).
Reviewer 2:
Tridegin is a folded peptide of 66 amino acids, with three disulfide bonds. It is an inhibitor of factor XIIIa, the transglutaminase that cross-links fibrin, and is of interest as a potential antithrombotic agent that does not activate thrombin either directly or indirectly. A 2-disulfide-bonded isomer of tridegin, B[C19S ,C25S] with disulfide bonds C5-C37 and C17-C31, is an active inhibitor and is the subject of this study. The structure of the peptide has been partially solved by NMR spectroscopic methods on two separate peptide fragments, N-terminal 1-37 and C-terminal 38-66, and the result subjected to molecular dynamics (MD), giving rise to a compact, folded structure that can be docked into the active site of fXIIIa.
Both of the disulfide bonds are in the N-terminal fragment, and the folded nature of this fragment gives rise to sufficient NMR-derived restraints to obtain a good, consistent structure. In contrast, the C-terminal fragment yielded no long-range restraints at all, and as a consequence the ‘NMR structures’ are extended in shape and indicate considerable flexibility. These are not true NMR structures but indeterminate structures, based on lack of data. It is only when a low energy NMR-based structure for the complete 66-mer is subjected to MD that the C-terminal fragment folds quickly into a compact conformation and remains stable for the remainder of the 300 ns MD run. The structure is therefore not entirely based on experimental data, as the final structure of the C-terminus, important for the activity of the peptide, owes very little to the NMR data, and indeed is probably not consistent with the NMR data observed. The authors should try to make this clearer, at least in the abstract; their structure is a combined NMR and MD structure.
Author comment: We thank the reviewer for his/her efforts dedicated to assess the manuscript as well as for the positive statement. We understand that it should have been described more precisely that the structure presented is a combined NMR and MD structure. Therefore, we have added a sentence that the structure of isomer B[C19S,C25S] is analyzed by a combination of NMR spectroscopy and MD simulation studies in the abstract (line 27, page 1) as well as in the end of the introduction (line 89,90, page 2).
One question arises – is the MD run reproducible? A single MD run might just be an artefact, but the same answer for two different runs could be taken more seriously. If possible, a second, confirmatory MD run should be undertaken.
Author comment: We agree with the reviewer’s question about the reproducibility of the MD simulation. We had considered this point earlier ourselves and had conducted a second independent MD simulation of the same system for up to 1000 ns. Analyzing the structures from the equilibrated trajectory from this 1000 ns confirmatory simulation revealed that the folding of the peptide was near-identical to the structure from the first 300 ns simulation. This confirms that the structure obtained from the 300 ns simulation is reliable and that the simulation had converged to produce a well-equilibrated structure. We have now added this information to the main text (line 230-233, page 9) as well as an additional supplementary figure (Figure S2) resulting from this analysis.
We very much hope that with the changes made and the corrections introduced you will find the manuscript improved and that it is now suitable for publication in the International Journal of Molecular Sciences.
Thank you for your kind attention.
Sincerely yours,
Diana Imhof

Reviewer 2 Report
Tridegin is a folded peptide of 66 amino acids, with three disulfide bonds. It is an inhibitor of factor XIIIa, the transglutaminase that cross-links fibrin, and is of interest as a potential antithrombotic agent that does not activate thrombin either directly or indirectly. A 2-disulfide-bonded isomer of tridegin, B[C19S ,C25S] with disulfide bonds C5-C37 and C17-C31, is an active inhibitor and is the subject of this study. The structure of the peptide has been partially solved by NMR spectroscopic methods on two separate peptide fragments, N-terminal 1-37 and C-terminal 38-66, and the result subjected to molecular dynamics (MD), giving rise to a compact, folded structure that can be docked into the active site of fXIIIa.
Both of the disulfide bonds are in the N-terminal fragment, and the folded nature of this fragment gives rise to sufficient NMR-derived restraints to obtain a good, consistent structure. In contrast, the C-terminal fragment yielded no long-range restraints at all, and as a consequence the ‘NMR structures’ are extended in shape and indicate considerable flexibility. These are not true NMR structures but indeterminate structures, based on lack of data. It is only when a low energy NMR-based structure for the complete 66-mer is subjected to MD that the C-terminal fragment folds quickly into a compact conformation and remains stable for the remainder of the 300 ns MD run. The structure is therefore not entirely based on experimental data, as the final structure of the C-terminus, important for the activity of the peptide, owes very little to the NMR data, and indeed is probably not consistent with the NMR data observed. The authors should try to make this clearer, at least in the abstract; their structure is a combined NMR and MD structure.
One question arises – is the MD run reproducible? A single MD run might just be an artefact, but the same answer for two different runs could be taken more seriously. If possible, a second, confirmatory MD run should be undertaken.
On the whole this is an interesting study, well written and presented, and represents a step forward in the study of tridegin, and the revisions needed are relatively minor.
Author Response

(The authors gave the same response as above.)
